# Direct Wafer-Scale CVD Graphene Growth under Platinum Thin-Films

**DOI:** 10.3390/ma15103723

**Published:** 2022-05-23

**Authors:** Yelena Hagendoorn, Gregory Pandraud, Sten Vollebregt, Bruno Morana, Pasqualina M. Sarro, Peter G. Steeneken

**Affiliations:** 1Laboratory of Electronic Components, Technology and Materials (ECTM), Department of Microelectronics, Delft University of Technology, 2628 CD Delft, The Netherlands; y.hagendoorn@tudelft.nl (Y.H.); g.pandraud-1@tudelft.nl (G.P.); s.vollebregt@tudelft.nl (S.V.); b.morana@tudelft.nl (B.M.); p.m.sarro@tudelft.nl (P.M.S.); 2Precision and Microsystems Engineering Department, Delft University of Technology, 2628 CD Delft, The Netherlands

**Keywords:** graphene synthesis, CVD, nanofabrication, thin films, silicon technology

## Abstract

Since the transfer process of graphene from a dedicated growth substrate to another substrate is prone to induce defects and contamination and can increase costs, there is a large interest in methods for growing graphene directly on silicon wafers. Here, we demonstrate the direct CVD growth of graphene on a SiO_2_ layer on a silicon wafer by employing a Pt thin film as catalyst. We pattern the platinum film, after which a CVD graphene layer is grown at the interface between the SiO_2_ and the Pt. After removing the Pt, Raman spectroscopy demonstrates the local growth of monolayer graphene on SiO_2_. By tuning the CVD process, we were able to fully cover 4-inch oxidized silicon wafers with transfer-free monolayer graphene, a result that is not easily obtained using other methods. By adding Ta structures, local graphene growth on SiO_2_ is selectively blocked, allowing the controlled graphene growth on areas selected by mask design.

## 1. Introduction

Since the discovery of graphene in 2004, it has become increasingly clear that the exceptional electronic, optical and mechanical properties of this material can lead to revolutionary new devices. Next generation transistors, RF circuits, optoelectronic devices and sensors with huge performance merits might become feasible. For producing such devices in high-volume and creating hybrid circuits comprising both graphene and silicon devices, it is highly desirable to integrate them with advanced complementary metal-oxide-semiconductor (CMOS) technology.

To realize this integration [1], production methods for fabricating graphene layers with dimensions equal to the size of a silicon wafer, up to 300 mm diameter, need to be developed. Moreover, it is required that the graphene can be deposited and patterned on silicon wafers without degrading its quality. After intense scientific and industrial research during the last decade, it now has become clear that chemical vapor deposition (CVD) is probably the most promising process to guarantee large-area high-quality graphene [2].

### 1.1. Transfer-Based Integration

In modern CVD processes, graphene is grown on foils, or thin films of transition metals, the most common of which are: Cu, Ni, Mo and Pt. For CMOS integration, the graphene needs to be transferred from these foils onto a silicon wafer. There are several ways to achieve this [3], but the procedure usually follows these steps: first a transfer polymer layer such as Poly Methyl Methacrylate (PMMA) is coated on the graphene, and the transition metal foil is selectively removed with a suitable etchant, after which the transfer polymer with graphene is placed on a silicon wafer (Figure 1a top). As the last step, graphene can be patterned using conventional lithography methods (Figure 1a bottom).

This transfer-based CVD graphene integration route has the advantage that the growth of graphene can be optimized completely independently from the substrate, enabling high-quality graphene to be integrated, without exposing the substrate to high temperatures. However, it also poses challenges, since the transfer procedure can result in imperfections in the CVD graphene due to (polymer) contamination, membrane fracture, wrinkling, and strain non-uniformities [4]. The grain boundaries in the polycrystalline transition metal foils or films that are used to grow the CVD graphene can also be a source of imperfections [5]. Although transfer-based CMOS integration is still the method of choice [2], these challenges have not been fully solved, such that actual device performance, and device-to-device uniformity continue to be limited by transfer related imperfections.

### 1.2. Transfer-Free Integration

Considering the aforementioned drawbacks of transfer-based graphene flows, it is of interest to investigate alternative ’transfer-free’ methodologies, where the graphene is grown directly on the substrate, such that the transfer process can be eliminated. However, the number of substrates on which graphene can be grown has appeared to be limited. Moreover, the high electrical conductivity and reflectivity of the transition metal substrates that are commonly used for graphene growth prohibit many device applications. A notable exception appeared to be silicon carbide, which allows direct growth of single-layer graphene on its surface and is still a strong contender for enabling graphene integration, despite the high price of crystalline silicon carbide wafers limiting production volumes.

An alternative route towards transfer-free graphene growth on silicon wafers is the use of a seed layer that is deposited on top of the substrate wafer for growing graphene. For such growth using a seed layer, the transition metals Cu, Ni, Mo and Pt, on which CVD graphene growth has been established, are logical choices. With Mo and Pt being the preferred seed materials, since Ni and Cu tend to form silicides or diffuse into the silicon faster.

During the last years, it was shown that few-layer CVD graphene can be grown on Mo seed layers [6] that are as thin as 50 nm (Figure 1b top). Moreover it was demonstrated that after graphene growth it was possible to remove (Figure 1b bottom) the seed layer by underetching [7,8,9]. This resulted in locally patterned, transfer-free graphene structures on the substrate that, after electrical contacting, function as gas-sensors [7] and pressure sensors [9,10]. This transfer-free growth of graphene certainly has a strong potential, in particular for micromechanical sensor applications, even though it does not offer very high electronic mobility and cannot be used for growing single monolayers of graphene.

In this work we explore another route toward transfer-free graphene integration by using Pt seed layers in a process flow where graphene grows beneath the seed layer, at the SiO_2_/Pt interface, as shown in Figure 1c. We are combining the benefits of CVD graphene growth on Pt with a transfer-free integration on silicon wafers. The process involves local graphene growth at the Pt/SiO_2_ interface, after which the Pt seed layer is removed. This process reduces organic contamination, mechanical stress and other potential challenges of transfer-based process flows. Moreover, we show that an additional intermediate, adhesion layer of thin film of Ta metal between Pt and SiO_2_ can locally inhibit the graphene growth with designed patterns. A similar study has been done by implementing a Ni thin film as a seed layer [11]; however, in that work the high carbon solubility of the metal was an impactful bottleneck towards control of the number of graphene layers.

### 1.3. Integration of Graphene Grown on Platinum

Platinum was one of the first transition metals that was used as a substrate for graphene CVD growth [12,13,14,15,16,17,18,19]. It follows from simulations that the distance between the graphene layer and the Pt substrate is 3.1 Å, which is significantly larger than that of CVD on Cu or Ni (2.24 Åand 2.01 Å) [14,20,21,22,23]. This large distance results in a small average adhesive energy per carbon atom (∼39 meV) [14,20], minimizing effects of the substrate on the graphene layer. Moreover Pt is more resistant to oxidation than Cu and Ni substrates. These properties have enabled the growth of large single-crystalline graphene islands on the platinum surface.

When graphene is grown on Cu or Ni foils, the foil is often etched away after the growth process. However, Pt foils are costly, and reuse of the foils is therefore often preferred, which has led to the development of processes to delaminate the graphene from the Pt surface [24], or grow it on much thinner Pt layers [25]. Growth on Pt can thus yield high-quality graphene, if the transfer challenges discussed in Section 1.1 can be dealt with. In the following, we discuss the methodology outlined in Figure 1c in detail and discuss the results and their implication and potential for integrated graphene-on-silicon device applications.

## 2. Methods

A 100 mm silicon wafer with a 90 nm thick film of thermally grown silicon dioxide is used as base substrate. On top of the SiO_2_, thin platinum PVD films with a thickness of 50 nm are evaporated from a Pt target (99.95% purity). The platinum films are annealed at 1000 °C under atmospheric conditions or in nitrogen gas at 1 bar for 60 min, resulting in a typical Pt grain size of ∼0.5–1.0 µm as observed from the SEM and AFM, see Figure 2. The peak-peak surface roughness was measured by AFM to be ∼15 nm for O_2_ and ∼12.6 nm for N. Since differences were small we will further on only present results using the nitrogen anneal. This pre-anneal was found to be essential to prevent dewetting of the Pt film during subsequent growth of graphene and overall improves the continuity of graphene deposition.

Graphene is deposited on these substrates using an Aixtron BlackMagic Pro cold-wall CVD system in a mixture of Ar and H_2_ gas at a pressure of 25 mbar, at 950 °C, while using methane gas as carbon atom precursor. Subsequently, the whole system is gradually heated up to 920 °C and kept stable for about 5 min before adding methane to the CVD chamber. For graphene growth, a CH_4_ gas flow of 30 sccm is applied for a duration of 150 s, after which the sample is cooled down. For ensuring high quality graphene growth, a fast cool-down of the sample at a rate of 50 °C/min was applied.

Besides the growth on blanket wafers, graphene was also grown on substrates with patterned Ta structures, as shown schematically in Figure 1c. These structures were made by sputtering 30 nm Ta film on the SiO_2_ surface and subsequent patterning by lift-off. Subsequently, a Pt thin film was evaporated, and graphene was grown using the same method described above. Ta was chosen for its high-temperature compatibility and because it withstands the aqua regia Pt etchant.

To assess whether graphene grows under the Pt seed layer (like in Figure 1c), the Pt thin film was removed completely by wet etching in a solution of hot (at 85 °C) Aqua regia (HCl (37%) 4:1 HNO_3_ (99%)) in 4 min and subsequently the wafer was rinsed in DI water and air-dried.

## 3. Results

The presence and the quality of graphene was investigated by a Renishaw inVia Reflex Raman spectrometer. The Raman microscope is equipped with a He-Ne laser (λ = 633 nm) and a 50× objective with a numerical aperture of 0.50, and is used in back-scattering configuration.

Figure 3 shows a typical Raman spectrum taken on the graphene after removal of the Pt. The peak ratios in this spectrum were found to be $I_{2D}/I_{G}=3$ and $I_{D}/I_{2D}=0.1$. These ratios correspond to monolayer graphene, which according to literature [26,27,28] has intensity ratios $I_{2D}/I_{G}>2$ and $I_{D}/I_{2D}<0.3$. By fitting the 2D peak by a Lorentzian a peak width (FWHM) of 44 cm^−1^ was found, close to that of monolayer CVD graphene [27] where FWHM = 38 cm^−1^ was found. Similar spectra were taken at different positions on the wafer, from which a mean integrated peak ratio $I_{2D}/I_{G}=2.25\pm 0.53$ was determined, indicating that most of the CVD graphene on the wafer was monolayer. Figure 4a shows an optical microscope image of a wafer on which 30 nm thick Ta blocking structures were patterned on SiO_2_ before Pt deposition and CVD graphene growth. Using Raman spectroscopy after Pt removal, it was established that graphene was only present on the SiO_2_ in between the Ta structures, and not on the Ta structures. A Raman spectrum of graphene on SiO_2_, with peak ratio and FWHM corresponding to monolayer graphene, is shown in Figure 4b.

## 4. Discussion

The basic mechanism [29,30] of interfacial graphene growth is likely not fundamentally different from previous CVD graphene growth studies on Pt, since it resembles the process observed [11] on Ni seed layers. Based on literature studies, let us propose a potential process that can lead to the observed interfacial graphene growth. At high temperatures, the methane gas is dissociated at the Pt surface into carbon and hydrogen atoms that are chemisorbed by the Pt catalyst [31,32,33,34,35]. The diffusion of carbon atoms through the Pt film proceeds mainly via the grain boundaries [36] and as a result of the small nanometer-scale thickness of the Pt layer, a uniform carbon concentration in the film will establish itself more rapidly than in Pt foils. Growth of graphene is facilitated at the SiO_2_/Pt interface, since the adhesion forces between these surfaces are weak because Pt does not form chemical bonds with the SiO_2_ surface. Possibly, the low surface roughness of the SiO_2_ will promote the flatness and quality of the graphene growth and the purity of the layer might be increased by the requirement for the carbon atoms to diffuse through the Pt. Although no Raman signs of graphene growth were found on top of the Pt layer, we cannot fully exclude its presence since the Raman signal of graphene on Pt is very weak [37]. We also note that graphene is not expected to grow at the SiO_2_/Ta interface because of the chemical bonds between Ta and SiO_2_, and the low diffusivity of carbon in Ta at the graphene growth temperature [38]. As additional evidence for the growth of graphene at the Pt/SiO_2_ interface, we present an optical micrograph of the wafers with patterned Ta after CVD graphene growth in Figure 5. The figure shows that the graphene layer grows below the Pt selectively in the regions without Ta, consistent with the picture sketched in Figure 1c. Figure 5 also indicates that the Pt in the regions without Ta adheres weakly to the SiO_2_, which might both be a cause and an effect of the CVD graphene layer growth between SiO_2_ and Pt. Besides acting as a blocking layer, the Ta metal might be used as electrode material. However, in the characterized samples it was observed that no electrical contact was formed between the graphene on SiO_2_ and the Ta metal. Additional process steps, or local etching of Pt might be used to establish electrical contacts.

## 5. Conclusions

We present a route for the direct growth of graphene on a silicon dioxide surface at the interface between Pt and SiO_2_. The high diffusivity of carbon in Pt, the high melting temperature and the high chemical inertness and catalytic activity of platinum metal make it ideally suited to promote the discussed growth mechanism. The ability to grow graphene directly on silicon substrates is clearly advantageous, since it circumvents the need for potentially detrimental graphene transfer steps. Moreover, we demonstrated selective graphene growth by using Ta thin films, which omits the need for graphene patterning steps. By process tuning, the quality of the graphene can probably be much further improved and integrated to provide a path towards high-quality graphene devices in application areas for sensing, data processing and high-speed communication.

## Figures and Tables

**Figure 1 materials-15-03723-f001:**
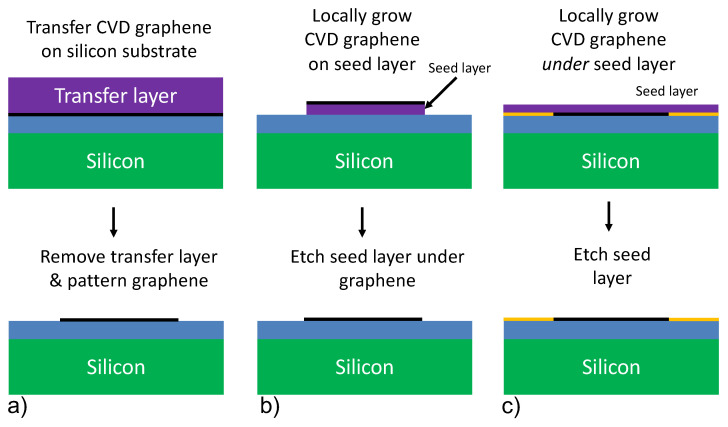
Schematic flows for transfer-based and transfer-free integration of CVD graphene on silicon wafers. (**a**) Transfer-based integration flow. (**b**) Transfer-free integration flow, growing graphene locally on top of a seed layer (purple). (**c**) The focus of this work: transfer-free integration flow growing graphene under a seed layer, where the yellow blocking structure layer is used to define the local graphene growth region. The blue layer in the figure can be the CMOS backend SiO_2_ layer, or any other interface of choice.

**Figure 2 materials-15-03723-f002:**
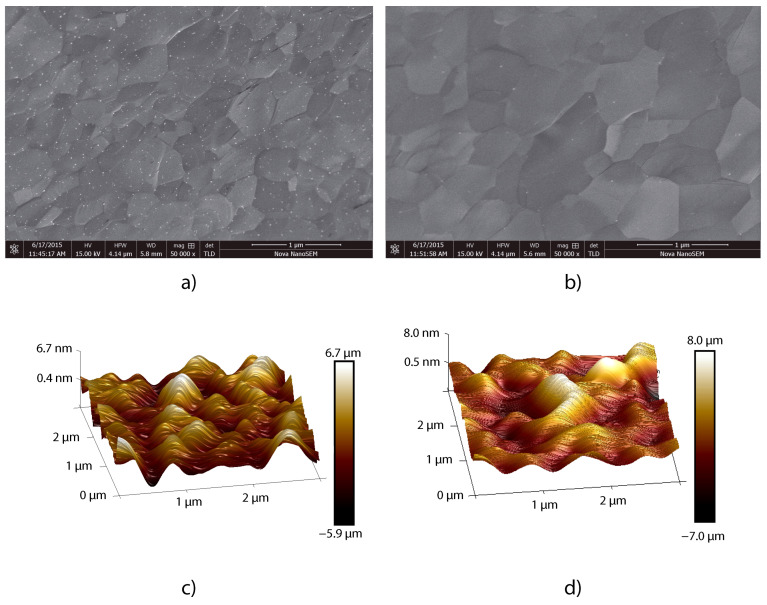
Scanning electron microscopy (SEM) and atomic force microscopy (AFM) of the platinum seed layer after annealing. (**a**) SEM of an area with a grain size of ∼0.5 µm after anneal in nitrogen gas. (**b**) SEM of an area with a grain size of ∼1.0 µm after anneal under atmospheric conditions. (**c**) Surface roughness as determined over an area of 3 × 3 µm^2^ by AFM after nitrogen anneal. (**d**) Surface roughness as determined by AFM after anneal under atmospheric conditions.

**Figure 3 materials-15-03723-f003:**
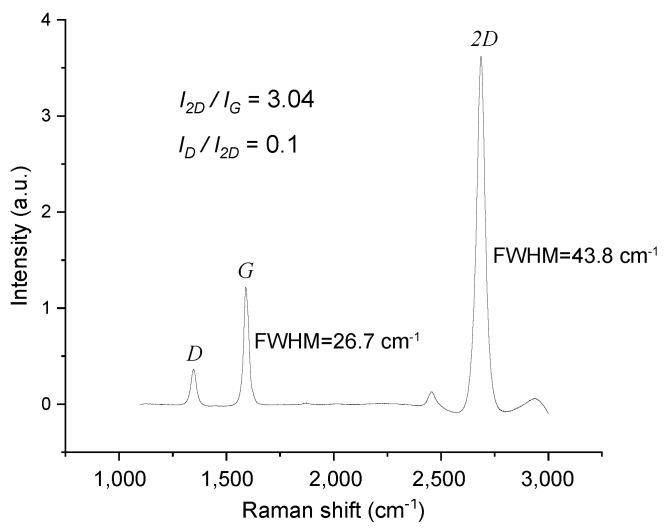
Raman spectrum of transferless CVD graphene on SiO_2_ after the removal of the Pt seed layer. The intensity ratio $I_{2D}/I_{G}=3$ between the 2D and *G* peaks at ∼2690 cm^−1^ and ∼1590 cm^−1^ is typical for monolayer CVD graphene. The full-width-at-half-maximum (FWHM) of the peaks is FWHM2D≈44 cm^−1^ and FWHMG≈27 cm^−1^ respectively.

**Figure 4 materials-15-03723-f004:**
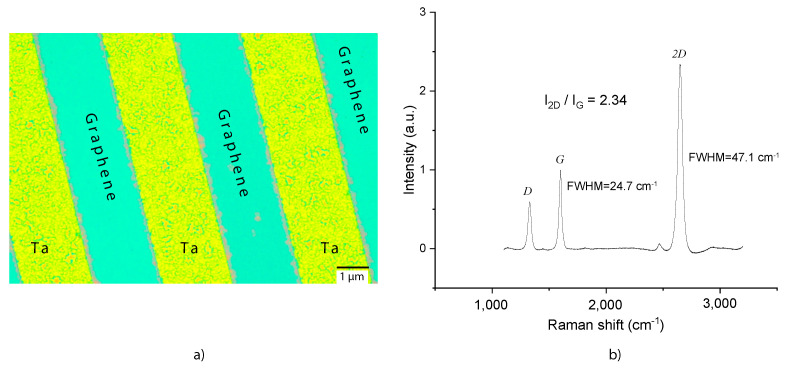
(**a**) Optical micrograph of a wafer with patterned Ta blocking structures (yellow) and transferless CVD graphene (turquoise). The graphene is grown at the interface between the SiO_2_ substrate and the Pt seed layer that is removed afterwards. (**b**) Raman spectrum of CVD graphene layer between Ta blocking structures, yielding Raman peak intensity ratios $I_{2D}/I_{G}=2.34$ and $I_{D}/I_{2D}=0.24$ and FWHM2D≈47 cm^−1^ representative for monolayer graphene.

**Figure 5 materials-15-03723-f005:**
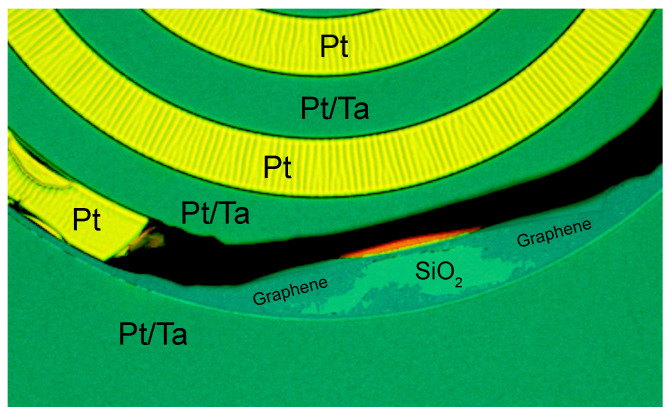
Optical micrograph of a wafer with lift-off patterned Ta structures, after the CVD graphene growth and before the Pt etch process step. Whereas the 50 nm Pt on top of 30 nm Ta is forming a flat surface, with Ta enhancing Pt adhesion, the intermediate regions with Pt show signs of weak adhesion (surface corrugation) and delamination. In a region where the Pt is delaminated, CVD graphene is observed on the SiO_2_, that corresponds to the graphene in Figure 4 after Pt etch.

## Data Availability

Not applicable.

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
