# Peer review of "Direct Wafer-Scale CVD Graphene Growth under Platinum Thin-Films"

_materials, 2022, doi:10.3390/ma15103723_

Round 1

Reviewer 1 Report

 The paper present a research on the growth of graphene
 directly on silicon wafers.  The main goal is to avoid
 imperfections and defects than can appear in the process of
 more traditional transfer methods.  The authors find than the
 use of Pt over previously use Ni (to allow the growth of graphene
 between the wafer and metal film) improves the yield of the
 method avoiding some problems previously found.

   The paper should be accepted, but I think readers will benefit
 from some other theoretical/experimental references where the
 transport in between metal film and wafer surface is discussed or
 investigated.

   A minor erratum, Ref. 11 is incorrect.

Author Response

We thank the reviewer for the useful feedback on our manuscript and its positive assessment for acceptance. Based on the comments we have added two references (Seah et al. and Sun et al.) that discuss the transport in between metal film and wafer surface. We also have corrected reference 11 (typing and page number).

Reviewer 2 Report

  • Authors should clearly explain the novelty of their work compared to similar works in this area.
  • What is the source of the D-band in synthesized materials? The CVD method is the route to synthesizing almost flawless structures. 

The authors should amplify the novelty of their finding and which potential application might be improved by their new method to make their paper be accepted in this journal.

Author Response

We thank the reviewer for carefully reading our manuscript and the useful feedback given. 

Comment 1: 

  • Authors should clearly explain the novelty of their work compared to similar works in this area. The authors should amplify the novelty of their finding and which potential application might be improved by their new method to make their paper be accepted in this journal.

Response 1: As suggested, we have explained the novelty of our work compared to similar works in the manuscript:

"In this work we explore another route toward transfer-free graphene integration by using Pt seed layers in a process flow where graphene grows beneath the seed layer, at the SiO2/Pt interface, as shown in Fig. 1c. We are combining the benefits of CVD graphene growth on Pt with a transfer-free integration on silicon wafers. The process involves local graphene growth at the Pt/SiO2 interface, after which the Pt seed layer is removed. This process reduces organic contamination, mechanical stress and other potential challenges of transfer-based process flows. Moreover, we show that an additional intermediate, adhesion layer of thin film of Ta metal between Pt and SiO2 can locally inhibit the graphene growth with designed patterns. A similar work has been done by implementing Ni thin film as a seed layer\cite{}, however, a high carbon solubility in the metal was an impactful bottleneck toward control of number of graphene layers."

The application potential of transfer-free graphene is described in the introduction, where we refer to references 1 and 2 that describe potential applications of integrated graphene on silicon. So we do not target a single application, but actually a large range of applications that require transfer-free graphene on silicon substrates. We would be happy to adapt further if there are specific points that require more clarification.

Comment 2: 

  • What is the source of the D-band in synthesized materials? The CVD method is the route to synthesizing almost flawless structures. 

Response 2: 

  • The presence of the Raman D-band is a sign of defects in the graphene, so it indicates that the CVD graphene that is grown is not perfect. Like indicated in the manuscript, typically the intensity ratio I_D/I_2D<0.3 for monolayer graphene. In this work we find  I_D/I_2D=0.1 which is within this typical range. We cannot answer the question in more detail based on the available experimental data and the complexity of the growth process. However, we strongly agree with the reviewer that it is of interest to improve the graphene quality (reducing I_D) and investigate the defect sources in the presented methodology in future work where we further optimise Pt catalyst morphology and CVD settings.

Reviewer 3 Report

In the manuscript “Direct Wafer-Scale CVD Graphene Growth under Platinum Thin-Films” the authors demonstrate the growth of a graphene oxide monolayer with a low defect content on a Pt/SiO2/Si substrate. The authors demonstrate the transfer-free mechanism for the formation of the Gr/SiO2/Si heterostructure. Under the assumption of the authors, the growth of graphene from the gas phase occurs at the Pt/SiO2 interface. Further removal of Pt slightly changes the defectiveness of the graphene layer. In my opinion, the main disadvantage of the work is the lack of evidence for the assumptions and the unconfirmed conclusions.

Based on the performed experiments and the presented data, there are a number of questions for the work:

1) How has it been proven that graphene grows on a Pt/SiO2 interface and not on a Pt/gas interface? There are works in the literature that indicate that the removal of the Cu or Pt layer can proceed without displacement of the graphene layer. In this case, there is a transition from the Gr/Pt/SiO2/Si structure to Gr/SiO2/Si. Note that the absence of a Raman signal is not a confirmation of the absence of graphene due to the strong interaction of graphene with platinum (DOI 10.1103/PhysRevB.88.235431).

2) The conditions of the experiment presented in the paper coincides with the experiments from the follow article “Monolayer graphene growth on sputtered thin film platinum (DOI: 10.1063/1.3254193)”. Why did the authors observe the growth of graphene on the upper face of platinum, while you observe it on the lower?

3) The article “Graphene Growth on and Transfer From Platinum Thin Films (DOI 10.1115/1.4038676)” shows that several effects occur during graphene growth on a Ta/Pt substrate. First, there is a redistribution of Ta and Pt, their homogenization, due to which the selective removal of one component is impossible. Second, the growth of a graphene monolayer can be observed on this substrate.

Author Response

We thank the reviewer for the careful study of our manuscript and the very relevant and useful questions. We address them here:

Question 1) How has it been proven that graphene grows on a Pt/SiO2 interface and not on a Pt/gas interface? There are works in the literature that indicate that the removal of the Cu or Pt layer can proceed without displacement of the graphene layer. In this case, there is a transition from the Gr/Pt/SiO2/Si structure to Gr/SiO2/Si. Note that the absence of a Raman signal is not a confirmation of the absence of graphene due to the strong interaction of graphene with platinum (DOI 10.1103/PhysRevB.88.235431).

Answer 1) We thank the reviewer for this important question. To provide evidence that the graphene is actually growing at the Pt/SiO2 interface, instead of being transferred from the top of Pt during the Pt etching process, we have added a Fig. 5 and 3 sentences to the discussion of the manuscript to explain Fig. 5 and its interpretation. This figure shows a delaminated Pt structure, and the presence of graphene below it. The presence of graphene below the Pt layer before the Pt etch shows that graphene growth is occurring at the Pt/SiO2 interface. We have also performed Raman on top of the Pt, and were not able to detect any graphene signatures within the SNR of our measurement, but we agree with the review that the Raman intensities on Pt are very low and hard to detect, such that this does not provide 100% evidence of the absence of graphene on top of Pt. To indicate this to the reader we have added a sentence to the manuscript: "Although no Raman signs of graphene growth were found on top of the Pt layer, we cannot fully exclude its presence since the Raman signal of graphene on Pt is very weak\cite{DOI 10.1103/PhysRevB.88.235431}.

Question 2) The conditions of the experiment presented in the paper coincides with the experiments from the follow article “Monolayer graphene growth on sputtered thin film platinum (DOI: 10.1063/1.3254193)”. Why did the authors observe the growth of graphene on the upper face of platinum, while you observe it on the lower?

Answer 2) An important difference is that the paper (DOI: 10.1063/1.3254193) uses a sputtered Pt film, whereas in our work we use an evaporated Pt film. As can be observed in the new Fig. 5, the adhesion of the evaporated film in our work to the SiO2 is very weak, which might facilitate the graphene growth at the Pt/SiO2 interface. The methane gas concentrations in the two works are also different. Furthermore we note that the quoted paper did not investigate the presence of graphene beneath the sputtered Pt film, so we cannot fully eliminate the possibility that graphene was also present below the sputtered film.

Comment 3) The article “Graphene Growth on and Transfer From Platinum Thin Films (DOI 10.1115/1.4038676)” shows that several effects occur during graphene growth on a Ta/Pt substrate. First, there is a redistribution of Ta and Pt, their homogenization, due to which the selective removal of one component is impossible. Second, the growth of a graphene monolayer can be observed on this substrate.

Answer 3) As discussed under question 1 we cannot fully eliminate the possibility of graphene growth on Ta/Pt and have added this possibility to the manuscript.

Round 2

Reviewer 2 Report

The authors responded to my concerns and the paper can be published in the journal.

Reviewer 3 Report

The publication satisfies the journal's requirements.